# Using Natural Language Explanations to Rescale Human Judgments

**Manya Wadhwa, Jifan Chen, Junyi Jessy Li, Greg Durrett**
Department of Computer Science
University of Texas, Austin
manya.wadhwa@utexas.edu

## Abstract

The rise of large language models (LLMs) has brought a critical need for high-quality human-labeled data, particularly for processes like human feedback and evaluation. A common practice is to label data via consensus annotation over human judgments. However, annotators' judgments for subjective tasks can differ in many ways: they may reflect different qualitative judgments about an example, and they may be mapped to a labeling scheme in different ways. We show that these nuances can be captured by *natural language explanations*, and propose a method to rescale ordinal annotations and explanations using LLMs. Specifically, we feed annotators' Likert ratings and corresponding explanations into an LLM and prompt it to produce a numeric score anchored in a scoring rubric. These scores should reflect the annotators' underlying assessments of the example. The rubric can be designed or modified after annotation, and include distinctions that may not have been known when the original error taxonomy was devised. We explore our technique in the context of rating system outputs for a document-grounded question answering task, where LLMs achieve near-human performance. Our method rescales the raw judgments without impacting agreement and brings the scores closer to human judgments grounded in the same scoring rubric.

## 1 Introduction

With the rise in model performance due to large language models (Brown et al., 2020; Ouyang et al., 2022, LLMs), there is a shift towards working on and evaluating more subjective tasks. On tasks like news summarization, LLMs can no longer reliably be judged by automatic metrics (Goyal et al., 2022a; Xu et al., 2023; Wang et al., 2023b) and achieve near-human performance (Zhang et al., 2023; Zhan et al., 2023). This high performance makes it harder than ever to annotate model outputs for errors (Saunders et al., 2022; Dou et al., 2022; Chang et al., 2023; Goyal et al., 2022b; Liu et al., 2023; Chen et al., 2024), an important ingredient as models are deployed in high-stake scenarios.

This shift towards more subjective tasks and difficulty in identifying errors has also led to a change in the role of human judgments in NLP. Earlier crowdsourcing work on simple labeling tasks explored capturing annotator bias with item-response models (Dawid & Skene, 1979; Smyth et al., 1994), especially to learn annotator quality (Hovy et al., 2013). As the ambit of NLP research has expanded to include more sophisticated downstream tasks, taking into account subjectivity (and thus, inherent disagreement) in

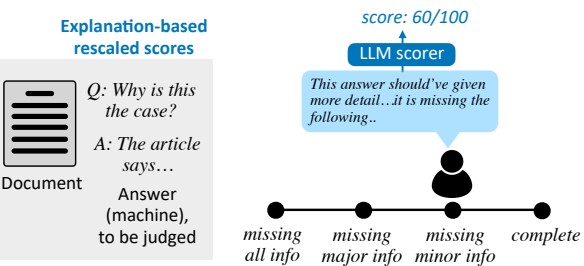

Figure 1: Overview of our method. By feeding explanations that annotators write into an LLM, we can rescale their coarse-grained judgment to a 100-point scale.

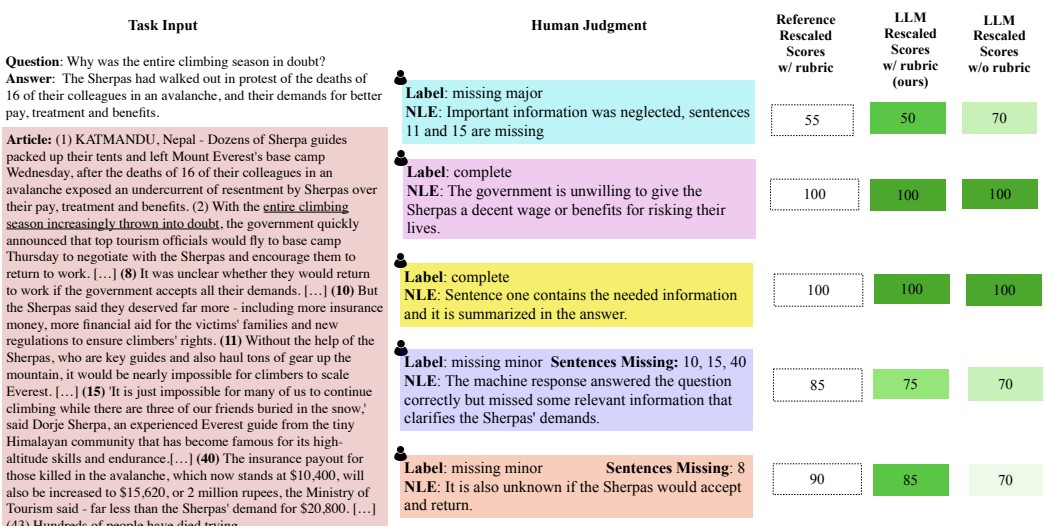

Figure 2: Example from INQUISITIVE-BROAD where annotators labeled a question-answer pair on a Likert scale and gave explanations for their judgments. Separate human labelers rescore these explanations according to an aspect specific scoring rubric. Our proposed explanation-based rescaling (EBR) method maps judgments to scores using that rubric, producing scores closer to the reference scores by accounting for factors mentioned in the explanations.

annotated data has surfaced as a key direction (Plank, 2022; Uma et al., 2021; Basile et al., 2021; Nguyen et al., 2016).

While existing work developed label aggregation methods and ways to incorporate different labels (Plank, 2022), information from the labels alone is still limited. Instead, intricacies in human judgments can be captured by **natural language explanations** provided during data annotation, which capture more nuanced subjectivity. They enable us to go beyond direct labeling, where annotators can choose different coarse labels for the same reason or the same coarse label for different reasons. The question is, *how do we transfer at least some of that expressiveness into numeric expressions that are more friendly for model training and evaluation?*

This paper proposes **explanation-based rescaling (EBR)** that enables us to transfer the expressiveness of natural language explanations into numeric forms. Our key idea is to make the ordinal label space (e.g., a coarse Likert scale) fine-grained, namely a 0-100 scale (Kocmi & Federmann, 2023), which then enables us to place the initial annotations in it by leveraging natural language explanations in a principled manner. Our approach begins by gathering explanations for each judgment during the annotation process. Next, we leverage an LLM to convert both a Likert judgment and its associated explanation into a numerical rating between 0-100. Crucially, this rescaling is guided by a predefined aspect-based scoring rubric, which can be defined entirely post-annotation, and provides task-specific guidance to the LLM on the placement of labels on the 0-100 scale. We find that an LLM with a rubric produces scores consistent with those that a human produces when using the same rubric.

We consider the task of evaluating LLM-produced answers for document-grounded, non-factoid question answering. We use annotated high-level questions (targeting discourse understanding) from the INQUISITIVE dataset (Ko et al., 2020), supplementing these with freshly collected questions from geographically diverse and recent text sources. We collect outputs from several LLMs, then annotate answers chiefly for *answer completeness*, with workers giving a Likert judgment and explanation. Through annotator recruitment and filtering, we selected a set of crowdworkers who gave high-quality, reasoned decisions, yet still had differences in opinion. Figure 2 gives an example from our dataset where annotators chose different Likert judgments for *information completeness*, but the explanations show that some of them were looking at the same factor. Rescaling with a scoring rubric is able to

impart fine-grained distinctions based on these explanations, and in particular, works better than rescaling without a rubric.

We evaluate our approach on whether LLM rescaling (1) can discern subtleties in natural language feedback as well as humans do; (2) changes correlation between annotators. The proposed approach brings scores closer to how humans would do this rescaling without impacting agreement, while retaining subjectivity.

Our main contributions are: (1) A method for rescaling Likert judgments with explanations using LLMs anchored in a scoring rubric. (2) A dataset of 12.6k human judgments (with explanations) on the completeness and correctness of answers produced by strong systems (GPT-3 and GPT-4) for document-grounded high-level question answering.

## 2 Background and Task Desiderata

**Motivating Example**  Figure 2 shows a motivating example for our approach on evaluating LLMs when they answer document-grounded questions. Given an article and a question, we use an LLM to generate a freeform answer. The answer to the question is high quality, but we want to be able to evaluate the LLM output precisely and assign granular scores. **Our approach is targeted towards tasks like this kind of LLM evaluation, where humans have to work hard to articulate subtle distinctions in their annotation.**

We have five crowd workers give judgments about the answer to this question.  They disagree about whether the question is missing major or minor information. For this task, it would be difficult to specify concrete enough annotation guidelines to make the label fully unambiguous. However, some of the explanations call out the same relevant feature of the answer, which is the answer was missing specific sentences talking about sherpas possibly not returning back.  **Our approach accounts for subjectivity in assessing quality, but assumes that there is an underlying ground truth answer which annotators can identify.**

An LLM can judge the provided labels and explanation, simultaneously giving finer-grained information ratings on a scale of 0-100. These ratings are anchored in a scoring rubric, which is defined post-hoc after looking at a collection of explanations for the aspect being evaluated (in our case it is for *completeness*). As mentioned above, the 0-100 scale offers granularity to effectively map a variety of explanations to numeric values, and the aspect-specific scoring rubric ensures consistency in this mapping, as opposed to mapping examples without on the numeric scale. We describe our method in detail in Section 5.

**Need for new data**  We need datasets that contain human judgments accompanied by informative, well-formed natural language explanations, where the underlying task satisfies the two properties bolded above (strong model performance and the right amount of subjectivity). We found three datasets that are relevant but they are not compatible with these characteristics. SQuALITY (Wang et al., 2022) includes evaluation of both human and model summaries by other humans. However, this task is very demanding due to the long document length, and we found that the feedback provided in the existing dataset is short and underspecified. Saunders et al. (2022) also have a dataset with human critiques of topic summaries. However, the evaluation criteria are defined loosely, leading to critiques that are too generic.  There is no multiply-annotated data for us to reliably perform analysis on the effect of our rescaling methods. Finally, Filighera et al. (2022) releases a dataset for short answer grading with explanations; however their data does not contain annotator information.  Note that other existing datasets for explanations in NLP, such as those mentioned in Wiegreffe & Marasovic (2021), do not align with our task requirements, since classification tasks do not involve fine-grained critiquing of outputs.

## 3 Collecting QA pairs

We study human judgments and critiques for non-factoid, document-grounded question answering, focusing on judgments of information completeness, i.e., whether machine-generated answers contain all key pieces of information from a document.  We use two

sources of data for this task. First, we use the questions collected in the INQUISITIVE dataset (Ko et al., 2020). Second, following the annotation guidelines from INQUISITIVE, a linguistics student wrote questions on articles focused on recent news events and non-western geographic entities (Hong Kong, Singapore, and India). This reduces leakage about notable prior news events for the LLMs in question. A summary of the two splits is given in Table 5. All text is in English. We call our dataset INQUISITIVE-BROAD.[1] See Appendix A for examples of articles for the task.

The questions target high-level comprehension of text, making the answers complex with information distributed across multiple sentences. They contain a mixture of causal questions (e.g., *why are they now liberalizing bank laws?*), procedural/elaboration questions (e.g., *how will the exceptions to the verification concept be taken care of?*), background information questions (e.g., *what is the main focus of this movement?*), and instantiation questions (e.g., *which groups were the human rights activists working on behalf of?*) (Ko et al., 2020), all of which require discourse-level processes and reasoning.

This task meets our desiderata by striking a balance between subjectivity and objectivity. There are divergent opinions about how much information should be included, but because the correct answer is grounded in the article, it should be relatively easy to judge if key information is missing. This contrasts with other long-form question answering tasks like ELI5 (**?**), where the lack of grounding makes it difficult to judge information completeness and requires subject matter experts for each question (Xu et al., 2023).

|  | Missing All | Missing Major | Missing Minor | Complete |
|---|---|---|---|---|
| Overall | 18 | 4 | 11 | 67 |
| text-davinci | 50 | 7 | 10 | 33 |
| text-davinci-003 | 8 | 3 | 12 | 77 |
| GPT-4 | 8 | 3 | 8 | 81 |
| EXPERT-HUMAN | 11 | 8 | 11 | 70 |

Table 1: Unaggregated % of labels in each category across all QA systems and for each system.

We use three off-the-shelf LLMs (Davinci, GPT-3.5-turbo and GPT-4) to answer questions from INQUISITIVE-BROAD. We also have two human experts answer a subset of the questions. These systems are shown in Table 6 and the prompts used are given in Figure 8.

## 4 Human Evaluation of QA pairs

Using Mechanical Turk, we enlist 8 qualified crowdworkers to conduct human evaluation of the question-answer pairs in our dataset. Each instance is evaluated by 5 crowdworkers. The study leads to a dataset of 12.6k annotations consisting of discrete labels along with natural language explanations.

Each crowdworker was given the article, question, and answer, and was asked to evaluate the answer on *completeness* and *correctness* attributes. For *completeness*, they were asked to choose on a Likert scale the amount of information missing in the answer with respect to all relevant information present in the document. Four options were given: *complete*, *missing minor* information, *missing major* information, and *missing all* information. For *correctness*, they were asked to mark if the information in the answer is faithful to the information present in the document. Crowdworkers were also asked to enumerate missing sentences and give rationale in the form of natural language explanation for their decision.

Appendix B.1 describes our process of recruiting and qualifying crowdworkers in more detail and shows screenshots of the interface (Figure 7). We ensure the crowdworker pay is ~$15/hour. Despite disagreements in annotation, we have substantial evidence, as well as third party validation (Appendix B.2), that these crowdworkers are attentive to the task and labeling high-quality data.

---

[1]The label distributions in the two data splits is similar, suggesting that leakage has no noticeable impact. Thus all results are reported on the entire INQUISITIVE-BROAD.

## 4.1 Dataset Statistics

Table 1 shows the un-aggregated % of labels in each category for the *completeness* attribute for INQUISITIVE-BROAD. The distribution is skewed towards the *complete* label. Table 1 also shows the distribution of labels for different QA systems being evaluated. Together, these show the challenges of judging the outputs of these systems. They are performing at a human level according to these annotators and make relatively infrequent mistakes.

Similarly, for the *correctness* attribute, 70% of QA pairs are marked as *correct*. Those marked as *incorrect* constitute of unanswerable questions where the model hallucinated information instead of accurately stating the information was missing from text. We thus focus our analysis on only the *completeness* attribute.

## 4.2 Human Label Variation

We also measure inter-annotator agreement across the discrete ratings. If we collapse to two classes, complete vs. not, the Fleiss Kappa value is 0.328. Kendall's Tau-b ($\tau_b$) correlation across all 4 labels is 0.325. This "fair" agreement, and given the perceived quality of the natural language explanations, we view this as evidence of genuine subjectivity.

Please refer to the Appendix (Table 10) for the distribution of labels across 3 annotators. Table 14 gives examples of explanations that differ in their label decision but agree on details in their explanations.

## 5 Explanation-Based Rescaling (EBR)

In this section we propose a rescaling method that captures natural language explanations by using them to re-evaluate the Likert labels. We take inspiration from the scoring mechanism presented in Kocmi & Federmann (2023) for machine translation evaluation, which validated the effectiveness of a 0-100 scale compared to other options. The key element here is that a chosen scale provides higher granularity than the coarse grained scale on which the initial Likert labels are collected. Instead of using an LLM to directly evaluate the task, we use a more detailed scoring rubric.

> **Rescaling Prompt**
>
> The main goal of your task is to score a machine generated response to a question. Scoring is on the "completeness" attribute. A complete answer will have all relevant information from the article required to answer the question and an incomplete answer won't. However, you are not directly scoring the machine response, but instead using a given feedback and amount of missing information. The details are given below:
> Article on which the question was asked: {article}
> Feedback given to the machine response: {feedback}
> Sentences marked as missing: {enumerated missing sentences}
> Level of the missing information:{label definition}
>
> On a scale of 0-100 how will you score machine response using the feedback and level of missing information stated above? Use the rubric below for scoring:
> 1. if the answer is complete, give 100 points
> 2. if the answer is missing one or more minor details then have deductions ranging from 5 to 30 points based on the severity of missing details
> 3. if the answer is missing a major facet of information, it results in a deduction of at least 40 points and more than 50 points are deducted if less than half of the correct information was given.
> 4. if the answer contains no correct information but only marginally relevant information from the article, 70 points are deducted
> 5. if the answer contains no correct information but the article clearly has information present, 100 points are deducted
>
> What is the score? Give a number.

Figure 3: The rescaling prompt for GPT-4, which considers the aspect definition (eg: *completeness*, defined in the prompt above), a scoring scale (0-100), the task input (the article), human judgment (Likert rating, natural language explanation and missing sentences) and a scoring rubric.

**Formalization** Assume we have a collection of items $x_1, \ldots, x_n$ which are being rated by annotators $a_1, \ldots, a_m$ in a sparse manner, i.e., not every annotator rates every item. Each annotator assigns an item a discrete rating $r_{ki}$ and writes an explanation $e_{ki}$, where $k$ refers to the item and $i$ refers to the annotator. We *rescale* annotator judgment as following: $s'_{ki} = f(r_{ki}, e_{ki})$.

Details about the rubric creation process are given in Appendix C. We also explore rescaling variations *without rubric*. Variants of the rescaling prompt are given in Appendix D.1 and we report our results in Section 7.1. Note, we do not rescale instances that get the extreme

labels *complete* and *missing all*, since in these cases we rarely observed nuanced explanations from which rescaling would be possible.

**Prompt**   To compute $f$, we invoke `gpt-4-0613`, taking as input both the explanation $e_{ki}$ along with the discrete rating label $r_{ki}$. The proposed rescaling prompt is given in Figure 3. The rescaling prompt is structured as following:

1. Aspect definition: defines the aspect being evaluated (in our case, *completeness*)
2. Scoring scale: defines the numeric scale to which the human judgment needs to be mapped (0-100 for our task; note that the LLM typically outputs multiples of 5)
3. Task input: helps contextualize the natural language feedback (the article)
4. Human judgment: Likert rating and the natural language explanation (and when available, missing sentences)
5. Scoring rubric with deductions: anchors the scoring to be consistent

# 6   Evaluation

With our initial human annotation and methodology established, we now turn to evaluation. We first introduce a notion of *reference rescaling*: given a rubric, can expert annotators agree on how explanations from our dataset should be rescaled? We establish data and baselines for this evaluation, then turn to a comparing human and automatic rescaling in Section 7.

## 6.1   Reference Rescaling

In order to evaluate the ability of LLMs to faithfully rescale natural language explanations, we ask three experts to establish reference scores. We sample 145 instances, where each instance consists of a QA pair and a corresponding human judgment in the form of the Likert label, natural language explanation and missing sentences. The distribution of labels in this subset is: *complete*: 20, *missing minor*: 52, *missing major*: 53, *missing all*: 20. We sample more from *missing minor* and *missing major* labels since these categories have nuanced explanations, compared to *complete* or *missing all*, which occur at extreme ends of the scoring scale. Each expert was given the same rubric information as the rescaling prompt mentioned in Section 5. We refer to an average of the expert rescaled scores as *reference scores*, represented by $R$.

Using this, we can answer two questions. (1) Can humans consistently rescale if given the rubric (i.e. is our methodology sound)? (2) Can LLMs do this rescaling *automatically* in a manner similar to humans when given the same rubric?

## 6.2   Metrics

We determine how well our rescaling method does by comparing the rescaled values with references scores and also by comparing inter-annotator agreement before and after rescaling. We use Mean Absolute Error (MAE) and Kendall's $\tau$ correlation (Kendall, 1948).

Mean Absolute Error computes a difference between $M(r_{ki}, e_{ki})$ and $R(r_{ki}, e_{ki})$, where $M$ is a system (e.g., our proposed rescaling method), $R$ refers to references scores (rescaling done by experts) and $(r_{ki}, e_{ki})$ is the Likert rating and explanation for item $x_k$ by annotator $a_i$.

Kendall's $\tau$ is based on pairwise score comparisons, and thus reflects a common ranking use case. We use Kendall's $\tau_b$, which makes adjustments for ties. Using a method $M$ (e.g., our proposed rescaling method), we compute correlations between the proposed and reference rescaled scores $\tau(M(r_{ki}, e_{ki}), R(r_{ki}, e_{ki}))$.

We also compute pairwise correlations between two annotators' rankings, which we denote as $\tau(M(r_{ki}, e_{ki}), M(r_{kj}, e_{kj}))$ for annotators $i$ and $j$, where $k$ is a placeholder index. We then compute an aggregate correlation across all pairs of annotators: $\frac{1}{\binom{|A|}{2}} \sum_{a_i, a_j \in A \times A, a_i \neq a_j} \tau(M(r_{ki}, e_{ki}), M(r_{kj}, e_{kj}))$, where $A$ is the set of annotators.

### 6.3 Baselines

We compare the proposed rescaling with four baselines that map human judgments to a numeric scale and calculate Kendall's $\tau$ and MAE against reference scores for each method.

STATIC rescaling maps the Likert ratings to a static numeric scale by assigning *complete* to 100, *missing minor* to 70, *missing major* to 30 and *missing all* to 0. This mapping does not use any fine-grained information from the explanations.

AVG EBR rescaling leverages the explanations to come up with a mapping for the four Likert labels to a 0-100 scale. To find this mapping, we first use EBR to rescale human judgment in the dataset. We then average all numeric scores under each of the labels. This method allows us to incorporate explanations while keeping the original level of granularity. This also maps labels at a calibrated interval from each other instead of having them at equal intervals (as in the STATIC rescaling method). We get the following mapping for each label: *complete*: 100, *missing minor*: 78.6, *missing major*: 50.9, *missing all*: 0.0.

EBR W/O RUBRIC rescales using human judgment which includes Likert ratings and explanations to get a score **without** the aspect-specific scoring rubric. This method can theoretically map examples with any Likert rating anywhere in the 0-100 numeric scale.

MISSING SENTENCES HEURISTIC (MSH) uses the number of missing sentences marked by the annotator as a way to rescale the original annotation. In this method, a deduction of 16 points is made for each sentence that is marked as missing; this value was set to minimize MAE. This scoring baseline is solely dependent on the number of missing sentences and does not incorporate any natural language explanations for evaluation.

|     |       | Missing Minor | Missing Major | Overall |
|-----|-------|---------------|---------------|---------|
| $\tau$ | (1,2) | 0.62[†] | 0.25[†] | 0.82[†] |
|     | (2,3) | 0.53[†] | 0.19 | 0.81[†] |
|     | (1,3) | 0.51[†] | 0.16 | 0.78[†] |
| MAE | (1,2) | 4.04 | 20.19 | 8.86 |
|     | (2,3) | 5.77 | 13.02 | 6.93 |
|     | (1,3) | 6.15 | 21.89 | 10.28 |

Table 2: Kendall's $\tau$ and MAE for expert rescaled scores. [†]: Statistically significant $\tau$ value with $p < 0.05$.

## 7 Results

In this section we show the effectiveness of our approach by comparing LLM rescaling against reference scores. We also explore how the rescaling influences inter-annotator agreement.

### 7.1 Rescaling Results

**Are humans able to do the proposed rescaling consistently?** To establish if humans can do the proposed rescaling we look at Kendall's $\tau$ and MAE between pairs of experts as reported in Table 2. Correlation is high and MAE is low when considering all labels. We also specifically look at *missing minor* and *missing major* labels; making distinctions within these requires the most nuance. Given the granularity of the proposed scoring scale (0-100) for our method, we see experts showing high correlation and low MAE for *missing minor*. However, for *miss-*

|  | Missing Minor | | Missing Major | | Overall | |
|--|---------------|---|---------------|---|---------|---|
|  | $\tau \uparrow$ | MAE $\downarrow$ | $\tau \uparrow$ | MAE $\downarrow$ | $\tau \uparrow$ | MAE $\downarrow$ |
| Without Rubric | | | | | | |
| Static | - | 15.1 | - | 12.8 | 0.85[†] | 10.1 |
| MSH | 0.49[†] | 12.5 | 0.35[†] | 25.4 | 0.42[†] | 24.2 |
| Avg EBR | - | 15.3 | - | 16.4 | 0.85[†] | 18.6 |
| EBR | 0.05 | 15.6 | 0.06 | 19.8 | 0.69[†] | 12.9 |
| With Rubric | | | | | | |
| Avg EBR | - | 8.7 | - | 14.1 | 0.85[†] | 8.4 |
| EBR (ours) | 0.45[†] | 7.9 | 0.20 | 14.4 | 0.83[†] | 8.1 |

Table 3: Comparison of the proposed rescaling against reference scores using Kendall's $\tau$ correlation and Mean Absolute Error. Reference scores are an average of three expert scores. EBR is the proposed method.[†]: Statistically significant $\tau$ value with $p < 0.05$

*ing major*, even though scores show low agreement we still see positive correlation on a granular scale. In Figure 4(a) we observe that one of the experts had higher spread of scores in *missing major* as compared to the other two, highlighting subjectivity in explanations and underscoring the complexity of this task. Table 12 in the Appendix shows examples of explanations given for the *missing major* label category.

**Does prompting GPT-4 with the rubric recover the rescaled values from humans?** Table 3 shows how our proposed rescaling using GPT-4 compares to reference scores on 145 instances mentioned in Section 6.1. We look at Kendall's $\tau$ and MAE between reference scores (average of expert rescaled scores) and EBR scores. We also compare EBR with the baseline methods defined in Section 6.3. Again, we specifically look at *missing minor* and *missing major* label categories, since they are the main source of subjectivity and have more nuanced explanations.

Our approach consistently outperforms all other methods in terms of achieving the lowest MAE, indicating its ability to faithfully capture subtle nuances in human explanations when compared to alternative rescaling techniques.

*Overall* correlation is consistently high across all methods. However, since this includes all labels, the *complete* and *missing all* categories can distort the overall metric. We instead choose to focus on *missing minor* and *missing major* specifically, looking at the correlation

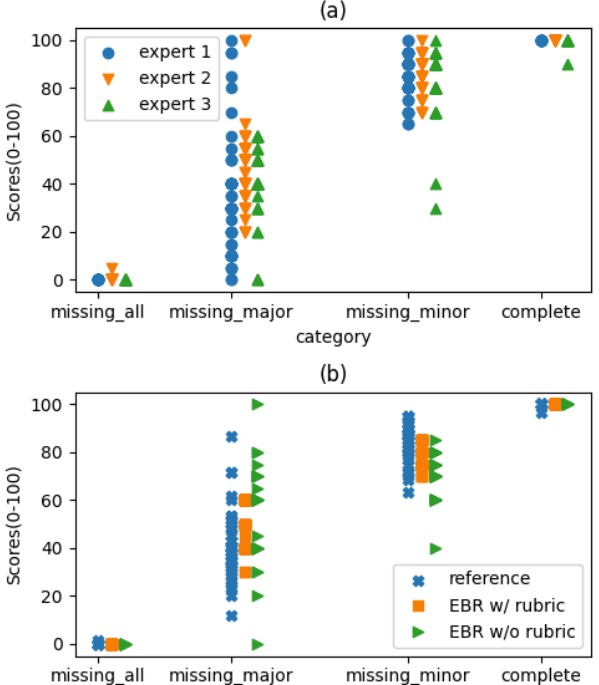

Figure 4: (a) shows the distribution of scores by the three experts. (b) shows the distribution of scores using EBR W/ RUBIC and EBR W/O RUBRIC against an average of the three expert scores (reference scores). EBR W/O RUBRIC is more scattered whereas the rubric causes the categories to track experts 2 and 3 more closely.

within these more subjective labels. Notably, with our proposed method both correlation values, i.e., 0.45 for *missing minor* and 0.20 for *missing major* are similar to the correlation values observed among experts, as shown in Table 2.

While MSH gets a higher correlation value than our baselines it also leads to a high MAE, even though this is the criterion it was optimized for. Note that STATIC and AVG EBR do not produce rankings within these categories, and therefore are not evaluated in this setting.

**How important is the rubric?** Table 3 shows the usefulness of the rubric. EBR with rubric consistently achieves lower MAE and higher correlation, across all labels and for *missing minor* and *missing major*.

In Figure 4(b), we also look at the distribution of EBR W/ RUBRIC scores and EBR W/O RUBRIC scores within each label category against reference scores. EBR W/ RUBRIC scores fall into label order without any explicit constraint. EBR W/O RUBRIC is more spread out and scores within a category cross boundaries with those of other label categories. Note that this is not necessarily wrong, but we view it as an indicator that the scores are less calibrated when no rubric is given.

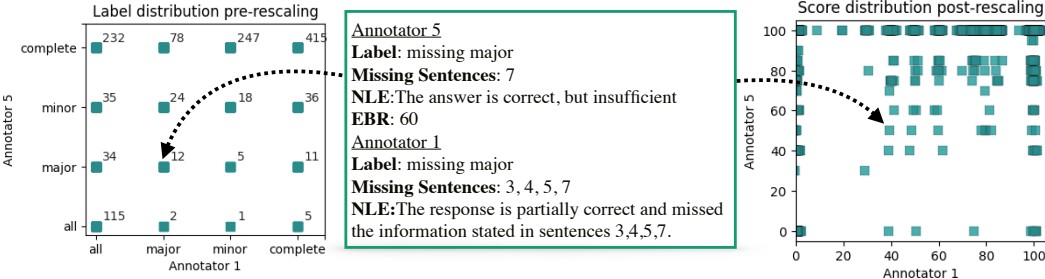

Figure 5: Class label distribution and post-rescaling model scores (EBR) for a pair of annotators. Note that horizontal jitter is added to differentiate points; all scores produced by our method are multiples of 5. The explanation-based rescaling imparts a finer granularity to the judgments while retaining subjectivity. We show a pair of explanations, where the two annotators agree on the label and but disagree on the finer details of what constitutes "missing major" information. EBR reflects the disagreement.

## 7.2 Annotator Alignment

Our proposed method transforms human judgment to a 0-100 scale, offering a finer level of granularity compared to the original four-category Likert scale used for coarse labeling. In this section, we investigate how going more fine-grained affects inter-annotator agreement.

**How does rescaling affect annotator alignment?** To understand how mapping to a fine-grained scale changes annotator correlation, we look at the average of pairwise Kendall's $\tau$ (as described in Section 6.2) for INQUISITIVE-BROAD.

Table 4 shows that our method is able to rescale human judgment to a much more fine-grained scale without impacting agreement; that is, our scoring can be more granular without impacting the inter-annotator correlation.

We take a closer look at a pair of annotators in Figure 5. It shows the distribution of labels (pre-rescaling) and scores (post-rescaling). For this pair there is no change in the agreement post-rescaling. Looking at an example of their annotations from the dataset, we see that they agree on the category label but identify different factors in their explanations. Our proposed method assigns different scores to each human judgment and is able to reflect this difference.

**How does rescaling impact subjectivity?** We emphasize that our goal is not to "smooth out" subjectivity in the annotation. Rather, taking natural language explanations into account produces a more calibrated and nuanced view that *preserves* inherent subjectivity while aligning differences between annotators when they actually agree.

Tables 13 and Tables 14 in Appendix G show examples of human judgments, which includes category label, explanation and missing sentences for two different questions. These show different styles of explanations, but with an overlap in missing information.

| Variation | $\tau$ |
|---|---|
| Original Labels | 0.33 |
| EBR without rubric | 0.32 |
| EBR (ours) | 0.32 |

Table 4: Average pairwise Kendall's $\tau$ for INQUISITIVE-BROAD. We look at how rescaling influences pairwise correlation with and without the rubric.

## 8 Related Work

**LLMs for evaluation and annotation** Kocmi & Federmann (2023) present a scoring mechanism for machine translation that we take inspiration from in this work. Other work has investigated giving prompts directly to LLMs instead of to annotators (Chiang & Lee, 2023; Wang et al., 2023a; Chen et al., 2023). Similarly, He et al. (2024); Gilardi et al. (2023); Törnberg

(2023); Zhu et al. (2023) showed that GPT-4 can outperform average crowd workers in several annotation tasks including BoolQ (Clark et al., 2019), Twitter content moderation (Alizadeh et al., 2022), sentiment analysis (Rosenthal et al., 2017), and more. Wang et al. (2023c) showed that GPT-3.5 can generate near expert-human instructions to align LLMs' behavior to instructions. Previous work has also explored the use of a defined criteria for model-based fine-grained error localization and rationale generation (Kim et al., 2024; Jiang et al., 2024). Our work is assumes that some tasks on the frontiers of LLM capabilities will always need human judgments; our procedure aims to augment and improve the capabilities of the group of crowdworkers.

**Building consensus among annotators** Another method for rescaling annotations is the *calibrated sigma* method (Weijters et al., 2016). However, this method does not change the ranking of examples, only rescales their scores for each annotator while preserving the ranking. We therefore do not compare to it here. Another line of work (Hovy et al., 2013; Gordon et al., 2021) focuses on modeling annotators to build better consensus judgments, but does not address aligning pairs of annotators. Sakaguchi & Van Durme (2018) proposed a method EASL, that adapts the original annotator labels to probabilistic ones based on the labels of all annotators. Finally, Ethayarajh & Jurafsky (2022) present a new protocol for NLG evaluation in which annotators give judgments in the form of probabilities over sets. This work also demonstrates the flaws in Likert judgments and pursues an approach to improving them orthogonal to ours. Consensus building has also been extensively studied in the translation community (Bojar et al., 2016). Graham et al. (2013) showed that using a continuous scale of 0-100 improved inter-annotator correlations on translation tasks.

**Natural language explanations** Natural language explanations (NLE) have been shown to be effective in improving model performance when used as additional features (Wu & Mooney, 2019; Murty et al., 2020; Liang et al., 2020), explaining model decisions (Camburu et al., 2018; Narang et al., 2020; Hase et al., 2020), boosting the performance of in-context learning of LLMs on a variety of reasoning tasks (Nye et al., 2021; Wei et al., 2022; Lampinen et al., 2022). Prior work also demonstrated that natural language explanations help us holistically understand annotator perspectives in complex tasks (Ferracane et al., 2021; Goyal et al., 2022a), improve annotation quality (Alonso, 2013) and aid adjudication (Filighera et al., 2022). Our work uses a similar idea but focuses on rescaling human annotation specifically, which is not a task addressed in this prior work.

## 9 Conclusion

In this work, we showed that LLMs can be used to rescale annotator judgments. We feed an annotator's label and explanation of that label into GPT-4 to produce a rubric-grounded score from 0-100. On a new dataset we collect of document-grounded questions answered by LLMs, we show that rescaled annotations align well with reference rescaled values produced by expert annotators. Overall annotator correlation does not change, but we show that our rescaling method is able to capture fine-grained nuances of the judgments, teasing apart subjectivity and scale use differences.

## Ethics Statement

Our work aims to broaden the role of human annotation in LLM development. By leveraging explanations of labels, we hope to use annotation in a more nuanced way and enable more effective human feedback. While this work focused on a document-grounded question answering setting, we envision that this can be useful for RLHF settings. Stronger feedback mechanisms like the one our work provides can lead to better aligned LLM systems and enable a wider range of (potentially less trained) annotators to steer this new technology.

## Acknowledgments

This work was partially supported by a grant from Open Philanthropy and NSF CAREER Awards IIS-2145280 and IIS-2145479. Thanks to Keziah Kaylyn Reina for annotating the extension of INQUISITIVE and conducting the worker quality analysis. Thanks to Keziah Kaylyn Reina, Kathryn Kazanas and Karim Villaescusa F. for rescaling 145 instances of human judgment and providing feedback on the proposed task.

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

# A  Data Collection

## A.1  Question Dataset

Table 5 shows statistics about INQUISITIVE-BROAD.

## A.2  Dataset Examples

Tables 7 and 8 show examples of articles, questions, system responses along with human annotations (ratings and explanations) for two articles from the INQUISITIVE-BROAD dataset.

| Property | INQUISITIVE | EXTENDED SET | Overall |
|---|---|---|---|
| num articles | 58 | 20 | 78 |
| avg sents/article | 37.1 | 34.6 | 36.2 |
| num questions | 358 | 231 | 589 |
| avg toks/question | 8.3 | 10.8 | 9.2 |
| num of annotations | 7058 | 5592 | 12650 |

Table 5: Dataset properties for two splits (questions taken from INQUISITIVE and our extended set) in INQUISITIVE-BROAD.

## A.3  Answer Collection

We prompt three LLMs and also ask two human experts (one of the co-authors and a non-author linguistic student) to answer questions from INQUISITIVE-BROAD. The systems along with their properties are mentioned in Table 6. EXPERT human in the table refers to linguistic undergraduate students who have worked on a series of computational linguistics annotation projects for 1-2 years. One student is now a lead data annotator at a small NLP company. These annotators are experienced at following annotation guidelines and producing high-quality judgments. Sample prompts for each of the systems is given in Figure 8.

| QA System | Property | Dataset Answered | Avg Answer Length (tokens) |
|---|---|---|---|
| Text-Davinci (davinci) | not instruct tuned | INQUISITIVE BROAD | 27.65 |
| Text-Davinci-003 (gpt-3.5-turbo) | instruct tuned | INQUISITIVE BROAD | 48.61 |
| GPT-4 (gpt-4) | instruct tuned + RLHF | INQUISITIVE BROAD | 48.54 |
| EXPERT human | - | INQUISITIVE EXTENDED | 50.40 |

Table 6: Systems used to answer INQUISTIVE-BROAD.

---

**Annotation Example - 1**

---

Article: CAIRO - A night of largely peaceful protests ended early Monday in a bloody clash between Muslim Brotherhood supporters and Egyptian soldiers, according to the Brotherhood and Egyptian media. Muslim Brotherhood officials, who are supporting ousted Islamist President Mohamed Morsi, said security forces raided their encampment outside the Republican Guard compound with tear gas and gunfire about 4 a.m. Supporters of Morsi have camped there for days demanding the release of the former leader, who has been under arrest since a military coup last week. Casualty figures were not immediately available, but Muslim Brotherhood officials said many people were killed and hundreds wounded. They called upon their supporters to donate blood and rush to the Nasr district of Cairo to assist the victims. 'Bloodbath!' tweeted Muslim Brotherhood spokesman Gehad Haddad. Egyptian television showed chaotic scenes of bloodied, unconscious protesters lying in makeshift triage facilities. They also showed images of more than a dozen bodies lying under sheets and Egyptian flags. In an interview with Al-Jazeera television, Haddad said Egypt had returned to a 'full-fledged police state in just five days'. Hours earlier, Egypt's new interim leadership had narrowed in on a compromise candidate to serve as the next prime minister. The state-run Ahram website and other Egyptian media reported that the new front-runner is Ziad Bahaa El-Din, a founding member of the Egyptian Social Democratic Party. El-Din is an attorney and former parliament member who previously served as an economic adviser, financial regulator and head of Egypt's General Authority for Investment under the government of deposed President Hosni Mubarak. El-Din is seen as a less divisive choice than secular opposition leader Mohamed ElBaradei, whose nomination was abruptly blocked a day earlier. Under the latest scenario, ElBaradei would become vice president, Ahram reported. El-Din and interim government officials cautioned that no final decision has been made. The debate over the prime minister marked the first standoff within the unlikely coalition of divergent opposition parties that came together last week to support the military's toppling of Morsi. The military and interim President Adly Mahmoud Mansour had hoped the swift appointment of a prime minister would send a message of stability and confidence to both Egyptians and the international community. Instead, the fumbling raised fresh doubts about Egypt's political future. Though viewed as a liberal, El-Din is seen as someone who will focus on improving Egypt's sagging economy and reforming corrupt government institutions. He has been an outspoken police critic and blamed Morsi for failing to adopt a more inclusive approach during his year in office. Liberal youth activists, led by the Rebel movement that helped organize the mass protests that ousted Morsi, pushed hard for ElBaradei, a Nobel Peace Prize laureate and former head of the International Atomic Energy Agency. Mansour seemed set to appoint ElBaradei, and a swearing-in was scheduled. But late Saturday the government abruptly backtracked after the ultra-religious Nour Party rejected him as too liberal and divisive, threatening to quit the interim coalition if ElBaradei got the job. On Sunday, throngs of Egyptians from both sides had voiced their views in mass protests - largely peaceful until the early morning violence. In the Nasr neighborhood, speakers denounced last week's coup as illegitimate and called on mid-level army officers to revolt against top generals who pushed Morsi out. 'We will stay here for as long as it takes and peacefully pressure the army to reinstate Morsi,' said Gamal Ragab, 41. And in Tahrir Square, the number of anti-Morsi demonstrators surged into the tens of thousands early Sunday evening. 'We want to build the democracy that we dreamed of in 2011, not a religious state where religion is used to divide the whole population and exclude anyone who doesn't share the Brotherhood's ideology,' said Samira Kamel, 21, a Cairo student. 'We want to show the world that this is a popular revolution, not a coup'.

---

**Question:** Does he have a lot of supporters?
**Davinci-003:** Yes, Muslim Brotherhood officials said security forces raided their encampment outside the Republican Guard compound with tear gas and gunfire about 4 a.m. and there were tens of thousands of anti-Morsi demonstrators in Tahrir Square.
**Annotator 1: Label:** Missing All , **Explanation:** The answer tells of anti-Morsi demonstrators not pro-Morsi. There is no information on the numbers of Morsi supporters
**Annotator 2: Label:** Missing Major , **Explanation:** The answer focuses on the anti-Morsi side, though the text only mentions "throngs" of protestors "from both sides"

---

Table 7: Example article, questions, system responses and annotations from INQUISITIVE-BROAD

| Annotation Example - 2 |
| --- |

Article: Touting her billionaire family 's legacy of populism and massive election victories , Thailand Paetongtarn Shinawatra is emerging as the candidate to beat in coming polls , betting that nostalgia can win millions of working class votes . Paetongtarn , 36 , is campaigning hard in the vote - rich rural strongholds of the Shinawatra family's Pheu Thai political juggernaut , hoping to reignite the kind of fervour that swept father Thaksin and aunt Yingluck to power in unprecedented landslides. Political neophyte Paetongtarn is promising Pheu Thai will complete unfinished business from three stints in office since 2001 , all of which were cut short by court rulings and military coups that it says were orchestrated by Thailand 's conservative establishment. " We managed to fix everything in the first year but then four years later we were ousted by a coup , so there are things that we have not achieved , " Paetongtarn said in her first formal interview with foreign media ahead of the election , expected in May . " So we go on each stage to tell people how our policies can change their lives. And only through stable politics can people's lives change in a sustainable manner , " she said , while campaigning in the northeast . Thaksin and Yingluck were toppled by the army in 2006 and 2014 , respectively , despite overseeing much economic growth. Both live in self - imposed exile to avoid prison convictions their allies say were designed to prevent their political comebacks. The baton has passed to Paetongtarn, Thaksin 's youngest daughter, who is using the same playbook in offering minimum wage increases , utilities subsidies , and long - promised high - speed rail systems and infrastructure to manage floods and droughts . Pheu Thai 's slogan is " Think Big , Act Smart " , taking aim at incremental reforms by the military - backed governments of Prime Minister Prayuth Chanocha since he seized power in 2014. " The picture has to be big and we must be able to address long - standing problems that festered. These must be completely dealt with , " Paetongtarn said . Though yet to be named as Pheu Thai 's prime ministerial candidate , Paetongtarn is far ahead in the opinion polls , with twice the support of Prayuth. Pheu Thai is expected to win most votes , but could struggle to lead a government given the military 's influence over an appointed Senate , which together with the elected lower house chooses the prime minister. Paetongtarn said she consults regularly and remains close with her father , who lives mainly in Dubai. His chief worry , she said , was her campaigning while nearly seven months pregnant . " But I'm OK , " she said. " This is my second pregnancy. I am aware of myself. I won't go too hard ." Despite their electoral popularity , the Shinawatras are loathed in Thailand as much as they are loved. They have long been accused by opponents of cronyism to enrich business friends and of buying off the poor with wasteful populist policies. The Shinawatras deny the charges. Thailand 's election is shaping up to be another grudge match between warring elites in Southeast Asia 's second - biggest economy. Paetongtarn said she remains concerned about the impact of the country 's intractable power struggle involving her family , including coups , which she said makes Thailand " go backwards ". " It also makes the world see our country in a different light. They don't want to trade with us. It reduces the opportunities for everyone , " she said. " Our country has been frozen for so long. So a coup should not take place again. The country must progress and people deserve to have better livelihoods ."

**Question:** How are foreign powers reacting to the election?
**Davinci-003:** Foreign powers have been cautious in their reactions to the upcoming election in Thailand. They are concerned about the country's intractable power struggle and the potential for further coups, which could have a negative impact on trade and other opportunities.
**Annotator 1: Label:** Complete , **Explanation:** The answer is correct and complete
**Annotator 2: Label:** Missing Minor , **Explanation:** The answer fits with the overall theme of the article, but it attributes Paetongtarn Shinawatra's concerns to foreign investors

Table 8: Example article, questions, system responses and annotations from INQUISITIVE-BROAD

# B    Human Evaluation of QA

## B.1    Annotation Task

We conducted four qualification rounds involving a group of 20 trusted and high-quality Turkers. These Turkers were required to meet specific criteria, including being located in the US, having a HIT approval rate of 98% or higher, and having completed more than 5000 approved HITs. They were assigned the task of annotating three documents, each containing five question-answer pairs. The label distribution varied across the documents. To ensure the Turkers understood and performed the task correctly, we manually reviewed the annotations and explanations. This rigorous review process aimed to confirm the Turkers' comprehension and execution of the task.

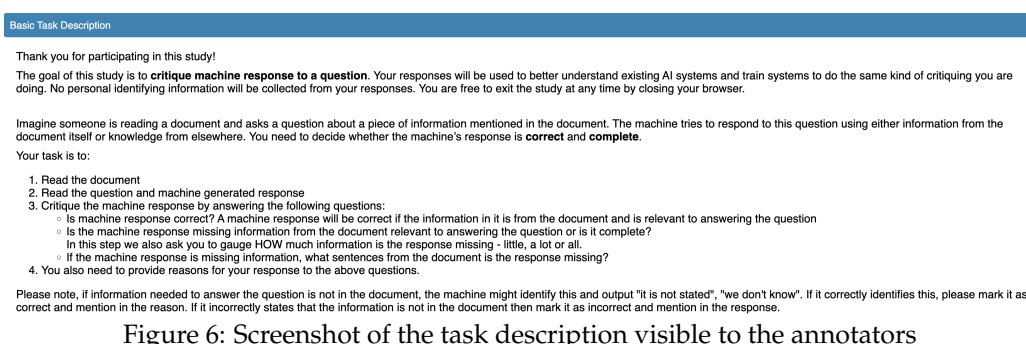

Figure 6: Screenshot of the task description visible to the annotators

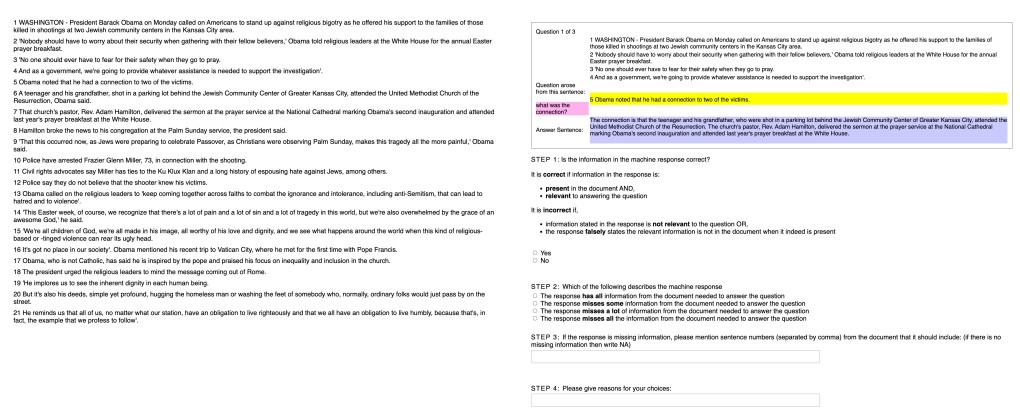

Figure 7: Screenshot of the annotation interface visible to the annotators

Following the qualification rounds, we selected 8 highly qualified Turkers. Given the substantial scale of our evaluation study, we actively monitored the feedback section of the HIT. If any ambiguities or issues arose, the Turkers had the option to contact us. Additionally, we promptly reached out to the Turkers if they provided any general feedback on the task, aiming to address and clarify any concerns.

In order to maintain quality of annotations, we assigned only one document per HIT. On average, each document had 4-5 questions. We compensated the Turkers $2 per HIT and gave them a 20% bonus after each batch. Each Turker got 2 hours to finish a HIT they accepted. We released batches of only 20 HITs every 12-24 hours. This aimed to prevent any decline in annotation quality that could potentially rise from prolonged or excessive workload.

Figure 7 shows a screenshot of the annotation interface visible to the annotators.

## B.2 Quality of the Explanations

In order to assess the quality of explanations we had a human expert (an independent judge not involved in this work) look at a sizable sample of explanations for each annotator in our dataset and mark if they find the explanations to be plausible. Specifically, the expert judge had access to the arti-

| Worker ID | 0 | 1 | 2 | 3 | 4 | 5 | 6 | 7 |
|---|---|---|---|---|---|---|---|---|
| % Quality agreement | 87 | 90 | 94 | 81 | 93 | 86 | 94 | 74 |

Table 9: Human evaluation of explanations to gauge worker quality.

cle and the QA instance, as well as the natural language explanation of the annotator;[2] the judge studies whether the explanations reflect sound underlying reasoning of the answer.

The expert evaluated 630 explanations in total. For each annotator, we sampled roughly 80 explanations, equally split across each label. Overall, our expert judge agreed with 87% of the explanations finding the granularity of the missing information to be plausible. Table 9 presents percentage of explanations with which our judge agreed for each annotator.

| Worker ID | missing all | missing major | missing minor | complete |
|---|---|---|---|---|
| 0 | 5% | 8% | 18% | 69% |
| 1 | 22% | 10% | 25% | 43% |
| 6 | 32% | 5% | 15% | 48% |

Table 10: Distribution of labels for three annotators for the INQUISITIVE question set with *davinci-003* answers. Worker 0 is more lenient whereas 1 and 6 are stricter.

Some sources of disagreement in the remaining 13% were 1) instead of critiquing the answer, annotators wrote the "correct" answers instead; 2) politically contested topics that led to the reasoning in explanations not being supported by the document; 3) differences (between the judge and the annotator) in reading of the question and differences in inference from what was provided in the document.

### B.3 Variation in Labels

Table 10 shows distribution of labels for three annotators. We see variations in annotator behavior, where annotator 0 is a little more lenient compared to annotator 1 and 6.

## C Rubric Creation

An important component of our proposed rescaling prompt in Section 5 is the aspect specific scoring rubric which is designed post-annotation. To create the proposed rubric, authors of this paper examined 20 human annotations, which included explanations and Likert labels. They assigned point deductions by taking into account factors that surfaced only with annotator's explanations after their annotation.

For instance, explanations under the label *missing minor*, frequently highlighted the absence of certain names in the response or the omission of multiple small details. Likewise, within the label category labeled as *missing major*, explanations commonly emphasized the role of the absent information in completing the response. Some explanations explicitly stated that "the response is lacking half the information" along with references to the specific sentence numbers that were missing. This information at a granular level provided the authors with valuable insights into the variations in errors within each label category, which further helped refine point deductions for the final rubric.

## D Prompting Details

### D.1 Prompt Variants for rescaling human judgment

The following is a variation of the prompt without the scoring rubric:

*The main goal of your task is to score a machine generated response to a question. Scoring is on the "completeness" attribute. A complete answer will have all relevant information from the article required to answer the question and an incomplete answer won't. However, you are not directly scoring the machine response, but instead using a given feedback and amount of missing information. The details are given below.*
*Article on which the question was asked: 'article'*
*Based on the above article, the following question was asked: 'question'*

---

[2]The judge did not see annotators' Likert ratings on the QA instances, since we believed that would shift their focus to the more subjective question of whether the explanation justifies the rating.

**GPT3-davinci**

article: New YorkCNN — Watching LVII in person will be cheaper than last year's game, but it will still cost you thousands of dollars. StubHub said that the average price customers are paying for a ticket Thursday was $6,800. The cheapest seats to Sunday's match up between the Philadelphia Eagles and Kansas City Chiefs are selling at about $3,200 apiece. Overall prices for the big game in Glendale, Arizona are trending down with prices dropping each day. In fact, StubHub said that ticket prices have dropped more than 10% since the teams were decided on February 12. Attending this year's game is costing fans substantially less than last year's match up between the Cincinnati Bengals and Los Angeles Rams at LA's SoFi Stadium. That's to be expected, since the home team was playing in their own city and the country's second-largest metropolitan area. Tickets were averaging nearly $10,000 each, StubHub. SeatGeek another ticket selling website, shows ticket prices for Sunday's game slowly declining and were averaging about $6,500 on Thursday. The most expensive ticket on its website costs $30,000 and is located near the field; the cheapest is $4,200. At an average of $445 per night, however, hotel prices in the Phoenix area will be relatively expensive, making it the second-highest level for a Super Bowl week, according to STR, a hospitality analytics firm that tracks prices. The most expensive Super Bowl was in 2016 in San Francisco when rooms averaged $451 per night. In addition to the Super Bowl, the city is also hosting a PGA event that features golf's biggest players, including Rory McIlroy and Jordan Spieth. Those two events on the same weekend puts the local hotel occupancy rate to 94%, the firm said. The normal rate for a hotel room is around $160 per night, STR said.
After reading sentence:'New YorkCNN — Watching Super Bowl LVII in person will be cheaper than last year 's game , but it will still cost you thousands of dollars.', a reader asked the following question.
Q:Why is the Super Bowl cheaper this year?
A:

**GPT3-davinci-003**

article: New YorkCNN — Watching LVII in person will be cheaper than last year's game, but it will still cost you thousands of dollars. StubHub said that the average price customers are paying for a ticket Thursday was $6,800. The cheapest seats to Sunday's match up between the Philadelphia Eagles and Kansas City Chiefs are selling at about $3,200 apiece. Overall prices for the big game in Glendale, Arizona are trending down with prices dropping each day. In fact, StubHub said that ticket prices have dropped more than 10% since the teams were decided on February 12. Attending this year's game is costing fans substantially less than last year's match up between the Cincinnati Bengals and Los Angeles Rams at LA's SoFi Stadium. That's to be expected, since the home team was playing in their own city and the country's second-largest metropolitan area. Tickets were averaging nearly $10,000 each, StubHub. SeatGeek another ticket selling website, shows ticket prices for Sunday's game slowly declining and were averaging about $6,500 on Thursday. The most expensive ticket on its website costs $30,000 and is located near the field; the cheapest is $4,200. At an average of $445 per night, however, hotel prices in the Phoenix area will be relatively expensive, making it the second-highest level for a Super Bowl week, according to STR, a hospitality analytics firm that tracks prices. The most expensive Super Bowl was in 2016 in San Francisco when rooms averaged $451 per night. In addition to the Super Bowl, the city is also hosting a PGA event that features golf's biggest players, including Rory McIlroy and Jordan Spieth. Those two events on the same weekend puts the local hotel occupancy rate to 94%, the firm said. The normal rate for a hotel room is around $160 per night, STR said.
After reading sentence:'Overall prices for the big game in Glendale , Arizona are trending down with prices dropping each day.', a reader asked the following question.
Q:How do these trends compare to other Super Bowl sales in previous years?
A:

**GPT-4**

While reading the article below:
New YorkCNN — Watching LVII in person will be cheaper than last year's game, but it will still cost you thousands of dollars. StubHub said that the average price customers are paying for a ticket Thursday was $6,800. The cheapest seats to Sunday's match up between the Philadelphia Eagles and Kansas City Chiefs are selling at about $3,200 apiece. Overall prices for the big game in Glendale, Arizona are trending down with prices dropping each day. In fact, StubHub said that ticket prices have dropped more than 10% since the teams were decided on February 12. Attending this year's game is costing fans substantially less than last year's match up between the Cincinnati Bengals and Los Angeles Rams at LA's SoFi Stadium. That's to be expected, since the home team was playing in their own city and the country's second-largest metropolitan area. Tickets were averaging nearly $10,000 each, StubHub. SeatGeek another ticket selling website, shows ticket prices for Sunday's game slowly declining and were averaging about $6,500 on Thursday. The most expensive ticket on its website costs $30,000 and is located near the field; the cheapest is $4,200. At an average of $445 per night, however, hotel prices in the Phoenix area will be relatively expensive, making it the second-highest level for a Super Bowl week, according to STR, a hospitality analytics firm that tracks prices. The most expensive Super Bowl was in 2016 in San Francisco when rooms averaged $451 per night. In addition to the Super Bowl, the city is also hosting a PGA event that features golf's biggest players, including Rory McIlroy and Jordan Spieth. Those two events on the same weekend puts the local hotel occupancy rate to 94%, the firm said. The normal rate for a hotel room is around $160 per night, STR said.
I had a question at sentence:'New YorkCNN — Watching Super Bowl LVII in person will be cheaper than last year 's game , but it will still cost you thousands of dollars.',
Why is the Super Bowl cheaper this year?

Figure 8: Examples of prompts for each QA system used to answer the INQUISITIVE-BROAD Dataset

*A machine responded with the following answer: 'answer'*
*Feedback given to the machine response: 'feedback'*
*Level of the missing information: 'label definition'*
*On a scale of 0-100 how will you score machine response using the feedback and level of missing information stated above? Give a number.*

It follows the same structure as the rescaling prompt in Figure 3 without the rubric.

### D.2 Cost of prompting

All models used in this work are accessed through OpenAI APIs[3]. The total cost of prompting LLMs in Table 6 for getting answers to INQUISITIVE-BROAD is ∼$30. All three models were prompted with the same set of articles and questions.

The total cost of prompting GPT-4 with EBR and the variations is ∼$500.[4]

## E  Average EBR scores per label

Table 11 shows the average score per label across all annotators using EBR for INQUISITIVE-BROAD. Although we do not impose any constraints regarding a score range per label when

---

[3]Pricing information for the models used can be found here.
[4]This includes multiple runs and prompt tuning on a small set of examples

applying this method to each example, the average of all scores for a label naturally aligns with the overall label order.

## F   Impact of Rescaling on Annotator Alignment

Figure 5 looks closely at a pair of annotators and their judgments. We see changes in label distribution pre rescaling and score distribution post rescaling.

| Label | Average Score |
|---|---|
| missing all | 0.0 |
| missing major | 50.0 |
| missing minor | 79.9 |
| complete | 100.0 |

Table 11: Average score of rescaled annotations for each original label across all annotators under the INQUISITIVE-BROAD dataset.

## G   Qualitative Impact of Rescaling

**NLEs with same scores**   Table 12 shows examples of explanations which got assigned the same score post-rescaling. We see some patterns in human judgment for these explanations. For example, *missing minor* explanations generally use phrases like 'somewhat correct', 'answered correctly, but missed some relevant information' where as *missing major* explanations tend to jump right into the information that was missed, along with mentioning sentences that were missing.

**NLEs for the same question**   Table 13 and Table 14 show human judgment as well as EBR and reference scores for two different questions.

## H   Stability of rescale prompting

To check the stability of our proposed method, we re-run rescaling four times on the 145 instances that also have expert rescaled scores. Table H shows the overall Kendall's $\tau$ and MAE for the different runs.

## I   Dataset Release

As mentioned in Section 3, INQUISTIVE-BROAD consists of two subsets of questions. The first set are questions collected in the IN-QUISITIVE dataset. Corresponding articles under this dataset are built on prior work which is sourced from Newsela and LDC. Newsela articles can be obtained from their website and LDC has standard purchasing guidelines. The second subset of questions is based on more recent events (March 2023). Due to copyright constraints, we are unable to disclose the processed article text. However, we will provide links to the original articles along with the code for processing them. Our question dataset will also be made publicly available.

|  | Avg Score | $\tau$ | MAE |
|---|---|---|---|
| Run 1 | 60.45 | 0.83[†] | 8.18 |
| Run 2 | 60.31 | 0.83[†] | 8.11 |
| Run 3 | 60.48 | 0.82[†] | 7.97 |
| Run 3 | 60.10 | 0.83 [†] | 8.16 |

Table 15: Results of running rescaling four times on the 145 expert rescaled instances. We look at Kendall's $\tau$ and MAE over four runs for the entire dataset.[†]: significant $\tau$ value with $p < 0.05$

We will be releasing all the data collected (questions, machine responses and human judgment) under the CC BY-NC license.

## J   Limitations

The effectiveness of our work hinges on the quality of the provided explanations and labels. Inconsistencies, such as the selection of incorrect labels by annotators (i.e., mistakes beyond

| | Label: Missing Minor, EBR: 85 |
|---|---|
| 1 | Explanation: The response is somewhat correct, but misses some relevant info
Missing Sentences: 9 |
| 2 | Explanation: The machine response answered the question correctly but missed some relevant information. It would be useful to include the fact that Argentinian passport holders can enter 171 countries visa-free since the machine response mentions Russians can enter only 87 countries visa-free. Including that fact in the machine response without additional context does would not make much sense.
Missing Sentences: 10 |
| 3 | Explanation: The response did not mention the street name as per sentence 5.
Missing Sentences: 5 |

| | Label: Missing Minor, EBR: 75 |
|---|---|
| 1 | Explanation: The machine response answered the question correctly but had its last sentence in its response cut off and missed relevant information. The response should include that the fake documents were issued to allow the women to settle in Argentina.
Missing Sentences: 23, |
| 2 | Explanation: The article doesn't provide a full answer to the question but contains some relevant detail.
Missing Sentences: 12, 29 |
| 3 | Explanation: Info from doc. sentence 3 is used to appropriately make the first sentence of the answer. The second sentence draws from doc. sentence 12. The final answer sentence is not relevant as the question is not asking how they are able to travel there. Info from sentence 17 which discusses fleeing the war and getting access to better health care.
Missing Sentences: 17 |

| | Label: Missing Major, EBR: 50 |
|---|---|
| 1 | Explanation: the machine response states what the helicopters were doing but fails to properly answer why they were operating (to combat rebel forces as indicated in sentences 15 and 35).
Missing Sentences: 15, 35 |
| 2 | Explanation: The machine response missed the question and provided an irrelevant response by restating the information in the source sentence. The article does not state exactly when horticulture was a priority for Americans but does mention an ongoing decline from peak membership in a horticultural association from the 1960s.
Missing Sentences: 24, 25 |
| 3 | Explanation: The response is partially correct and failed to mention the domestic factors as per sentences 4,7,8,13,14.
Missing Sentences: 4, 7, 8, 13, 14 |

| | Label: Missing Major, EBR: 40 |
|---|---|
| 1 | Explanation: the machine response missed the question and provided a mostly irrelevant response. line 27 explains that david hockney is 'one of the living masters of oil painting' and that was not present in the response explaining why hockey is britain's most celebrated living artist. additionally, the machine response conflates information presented in the article. the variety of mediums was taken from line 6 but hockney did not create all of those works as explained on line 32. |
| 2 | Explanation: The text does not provide a specific answer, but a lot of detail could have been included in an attempt to address the question.
Missing Sentences: 17, 18, 30 |
| 3 | Explanation: the answer makes use of sentence 1, but misses the specific people named in 3, 6, and 13:sergei grigoryants, vladimir oivin, marina shemakhanskaya. |

Table 12: We give examples of explanations that get rescaled to the same score. These explanations are for different questions and by different annotators. All the explanations are fine-grained, pointing sentences that were missing from the machine response, along with the level of severity of this missing information.

subjectivity), may pose challenges that are difficult to overcome. Additionally, further investigation is needed to determine the specific types of explanations that should be sought from annotators to facilitate more faithful rescaling.

Our proposed technique relies on a carefully curated scoring rubric. As outlined in Section 5, this process involves post-annotation analysis of the human judgment, which can be

| | Why is inflation expected to remain elevated? |
|---|---|
| 1 | Label: missing major
Explanation: The response is partially correct as it missed other factors as per sentences 8,12,13 and14.
Missing Sentences: 8, 12, 13, 14
EBR: 40
Reference score: 35 |
| 2 | Label: missing minor
Explanation: This is a comprehensive answer, but it was cut off before it could finish supplying key information
Missing Sentences: 8, 10
EBR: 80
Reference score: 80 |
| 3 | Label: missing major
Explanation: The first sentence in the response is incorrect as per sentences 1 and 2. While the rest of the response is correct, it missed the other factors mentioned in sentences 8,12,13 and 14.
Missing Sentences: 1, 2, 8, 12, 13
EBR: 30
Reference score: 26.67 |
| 4 | Label: missing major
Explanation: The first two answer sentences are not relevant as they are stating things that are inflated in price rather than why they are that way. The rest of the answer lists accurate reasons but misses global energy prices being higher (sentence 4), strong housing demand and tight entitlement quotas (sentence 8), and persistent manpower shortages (14).
Missing Sentences: 4, 8, 14
EBR: 40
Reference score: 30 |
| 5 | Label: missing major
Explanation: The response is partially correct and failed to mention the domestic factors as per sentences 4,7,8,13,14.
Missing Sentences: 4, 7, 8, 13
EBR: 50
Reference score: 23.33 |

Table 13: Human judgment along with EBR and reference scores for the question *Why is inflation expected to remain elevated?*

time-consuming, although this kind of "prompt engineering" is a fixed cost regardless of dataset size. We consider the discovery of such rubrics from human judgments as a potential avenue for future research.

Our analysis is also constrained by limited scale. Acquiring human annotations is expensive, which is why our study is restricted to one dataset and one aspect. However, we believe that the proposed method can be extended to various aspects and tasks that necessitate nuanced evaluation. Our method provides a template for future work, particularly if task designers are sensitive to the requirements of explanations for our method and infuse this understanding into the annotation process itself.

| How long did it take to end? |
|---|
| 1 | Label: missing major
Explanation: The response isn't useful and misses important information that could better answer the question.
Missing Sentences: 16, 17, 21, 22, 29, 30
EBR: 40
Reference score: 25 |
| 2 | Label: missing minor
Explanation: The answer is pretty good. It is relevant and includes information from the document, but fails to mention some of the details and other species that are still not recovered.
Missing Sentences: 11, 17, 18, 19, 34, 43
EBR: 70
Reference score: 68.33 |
| 3 | Label: missing major
Explanation: The text does not provide a specific answer, but a lot of detail could have been included in an attempt to address the question
Missing Sentences: 17, 18, 30
EBR: 40
Reference score: 48.33 |
| 4 | Label: missing minor
Explanation: Some more detail could have been included.
Missing Sentences: 46, 48, 50
EBR: 75
Reference scores: 63.33 |
| 5 | Label: missing minor
Explanation: Some additional relevant information was included in the article
Missing Sentences: 19, 21, 25, 46
EBR: 70
Reference score: 75 |

Table 14: Human judgment along with EBR and reference scores for the question *How long did it take to end?*

