# OpenReview forum: "Using Natural Language Explanations to Rescale Human Judgments"
_colmweb.org/COLM/2024/Conference — COLM_

### Official Review · Reviewer_8NAr · 2024-05-08

**Rating:** 5
**Confidence:** 5
**Ethics Flag:** 1

**Summary:**

This paper presents a method of using LLM to convert human Likert ratings to scalar values using the human explanations.
They first collected the Inquisitive-board dataset. The dataset consists of news articles and questions written by a linguistics student, and answers from LLMs. Then they ask annotators to evaluate the generated answers on how much information (major/minor/none) from the original articles the answers are missing, provide an explanation and how many sentences are missing. They found some disagreement in the Likert scale (major/minor/complete) ratings

They then provide some rubrics for converting the Likert scale ratings and explanations to a score in 0-100 for how much information is missing. They have 3 experts doing this rescaling for reference, and evaluate how well LLMs can do this rescaling. They found that the LLM rescaled values correlate with expert rescaled values and explanations provide fine-grained information beyond the Likert scale.

**Questions To Authors:**

I suspect the extent of information missing probably also correlate with the location of the missing sentence, or more broadly the information structure. Generally the information in the beginning of the article is probably more prominent and major than the information at the end of the article. The question being asked will also impact which information is prominent. Have you looked at this in your analysis?

**Reasons To Accept:**

This paper provides an interesting method on rescaling Likert scale judgments and showed that the explanations provide insights into the discrete judgments.

The methodology is sound and the approach is interesting.

**Reasons To Reject:**

The biggest question I’m left with after reading this paper is: what does one do with this method? The amount of missing information is a limited dimension of a generated answer. Can this method be extended to reconcile multiple and potentially conflicting dimensions? Since this paper is in the evaluation track, do the authors recommend using this method for future LLM evaluation? Does using the rescaled values as an evaluation metric leads to different models being preferred and are those models indeed better in some ways? Without clear takeaways/recommendations the impact of this paper is likely limited.

---

> ### Author Rebuttal · Authors · 2024-05-29
>
> Thank you for your review! Addressing your concerns below:
>
> > what does one do with this method?
>
> One use case is LLM evaluation as you say. This can be with human or model generated feedback where directly scoring the generation is subjective and we need natural language feedback to tease out nuances. Recently, rubric based scoring is being adopted in NLP [1] [2], but it work assumes a static rubric and produces the feedback as an explanation or a rationale to the score. Our work focuses on using a rubric defined post-hoc, looking for patterns in the feedback so that the evaluation can be adapted to the underlying subjective judgment.
>
> Another application could be fine-grained reward modeling. We see emerging frameworks like fine-grained RLHF [3] that provide ways to use fine-grained annotation scores. In line with that framework, we can evaluate the benefits of our proposed work to train better aspect-specific reward models. This can be further used to improve tasks like document-grounded QA, query focused summarization, or any task that involves human-LLM interaction.
>
> We note that these are emerging research directions themselves and there are not “plug-and-play” applications for the evaluation methodology we’ve developed here. We believe it can be extended in current work and conceptually helps in many possible ways, but it is not easy to connect to one of these downstream applications within the scope of this current paper.
>
> > Can this method be extended to reconcile multiple [..] conflicting dimensions?
>
> Our method is tied to the scoring rubric and the quality of explanations, so more exploration would need to be done to see whether or not we can reconcile multiple dimensions in the rubric. This can definitely be an interesting future direction we explore.
>
> > I suspect the extent of information missing
>
> This is an interesting hypothesis. In general, the information answering the questions can be located anywhere in the article, so we don’t strongly observe this. It would be interesting to connect more general notions of information relevance with these question answers, but that’s beyond the scope of the current work.
>
>
> [1] UltraFeedback: Boosting Language Models with High-quality Feedback https://arxiv.org/abs/2310.01377
>
> [2] Prometheus: Inducing Fine-grained Evaluation Capability in Language Models https://arxiv.org/abs/2310.08491
>
> [3] Fine-Grained Human Feedback Gives Better Rewards for Language Model Training https://arxiv.org/abs/2306.01693

---

### Official Review · Reviewer_f43i · 2024-05-09

**Rating:** 6
**Confidence:** 3
**Ethics Flag:** 1

**Summary:**

The paper seeks a way to improve human evaluations of subjective tasks via large language models (LLMs). In particular, the authors exploit the natural language explanations to rescale ordinal annotations. The authors propose a method called Explanation-Based Rescaling (EBR) that utilizes natural language explanations provided by human annotators alongside their Likert scale ratings. EBR leverages an LLM to convert both the rating and explanation into a more fine-grained numeric score (0-100) guided by a post-hoc defined scoring rubric. The authors demonstrate the effectiveness of EBR on a document-grounded question answering task where LLMs achieve near-human performance. They show that EBR produces scores closer to reference scores generated by experts using the same rubric, without impacting inter-annotator agreement.

**Reasons To Accept:**

S1. Utilizing natural language explanations to refine and quantify human judgments is novel. EBR allows for a more fine-grained evaluation of LLM outputs by mapping them to a 0-100 scale. This is particularly useful for tasks with subtle distinctions where traditional Likert scales might be too coarse. Furthermore, its flexibility (the fact that the scoring rubric can be designed post-annotation) can give room for better optimization.

S2. The authors conduct thorough experiments and analysis on a newly collected dataset, demonstrating the effectiveness of EBR compared to various baselines and highlighting its ability to capture subtle nuances in human explanations.

**Reasons To Reject:**

W1. Achieving high agreement among human evaluators for subjective tasks is a significant challenge. As shown in Table 2, reaching a consensus among human annotators on missing majors seems to be particularly difficult. Given this, using the results of human annotators as references may not be the most effective method for evaluating the capabilities of LLMs on this task. While I don’t have a clear solution for this issue, it’s certainly a challenging area that requires further exploration.

W2. It was unclear whether fine-grained labels are necessary and how they could be helpful in improving LLMs or for NLP overall.

W3. Regarding the evaluation, the authors mention that the work aims to demonstrate the usefulness of using NLEs to enhance rescaling performances. To achieve this, the experiment should be designed to allow for a comparison between sets with and without explanations. The current setup does not directly evaluate this effect.

W4. The evaluation is centered on a specific task of document-grounded question answering. Further investigation is needed to assess the generalizability of EBR to other tasks and domains. Specifically, in the presented task, the quality of the answers can be determined based on the missing sentences, which are relatively straightforward to quantify. However, in other scenarios, such clear criteria may not exist.

W5. Lastly, the paper acknowledges that the creation of the rubric is currently a manual process. This could be time-consuming and subjective, potentially affecting the generalizability of the approach.

---

> ### Author Rebuttal · Authors · 2024-05-29
>
> Thank you for your reviews and detailed comments! Addressing your concerns below:
>
> > Achieving high agreement among [...] requires further exploration.
>
> Our main goal for using human annotators as reference was to evaluate whether or not LLMs can do our rescaling task according to the instruction (or task rubric). We definitely agree with you that the task is challenging and this is reflected in our experiments as well (in Table 2). We are exploring some of these challenges in our future work.
>
>
> > It was unclear whether fine-grained labels are necessary and how they could be helpful in improving LLMs or for NLP overall
>
> In the era where LLMs make subtle mistakes and tasks are getting more subjective, we believe fine-grained scores are needed for evaluation. Fine-grained scales can give us a more expressive mechanism for representing differences in opinion where errors can be subtle: for instance, three distinct error types can be mapped to -5/-10/-15, all reflecting that a response is still quite good but at different levels. Note that our scores are all multiples of 5, so in reality our scale has 21 distinct values that are output from the model. A fine-grained scale can be coarsened if needed (eg: from 0-100 -> 1-4).
>
> That said, collecting human annotations on a fine-grained scale might be harder and there can be scale usage differences, which is why we suggest a method that goes from explanations to fine-grained scores.
>
>
> > Regarding the evaluation, the authors mention that the work aims to demonstrate the usefulness of using NLEs to enhance rescaling performances.
>
> We have comparisons to rescaling without explanations, reported in Table 3. In our proposed baselines, *Static* rescaling and *MSH* (Missing Sentences Heuristics) rescaling do not use explanations. We show that our proposed method that uses the explanation does substantially better compared to the non-explanation baselines.

---

### Official Review · Reviewer_skN9 · 2024-05-11

**Rating:** 3
**Confidence:** 4
**Ethics Flag:** 1

**Summary:**

This submission proposes a method to rescale coarse numerical ratings to fine-grained ones. The envisioned scenario is that for a task like document-based question answering, annotators asked to rate answers have given coarse-grained ratings on a four-point Likert scale ("complete", "missing minor", "missing major", "missing all"), along with short free text justifications of their rating. Noting the inherent subjectivity of these ratings and justifications, the authors assume that a finer-grained numerical scale from 0 to 100 points is more appropriate. To rescale from coarse to fine ratings, the authors prompt LLMs with the rating, justification, and an instruction that describes how to assign scores.

The evaluation of the rescaled scores focuses on the two middle labels ("missing minor" and "missing major") since the extreme points of the scale ("complete", "missing all") show little variation among raters. The evaluation is mainly performed by measuring agreement of LLM-rescaled scores with a rescaling performed by three experts on 145 instances. Agreement is measured via rank correlation (Kendall's Tau) and mean absolute error (MAE). In terms of rank correlation, the proposed method fails to outperform one of the simple baselines, namely the "missing sentence heuristic". In terms of MAE, the proposed method outpeforms all baselines, but unlike for rank correlation, there are no significance scores given, which is a concern considering the small sample size of 145 instances.

**Questions To Authors:**

- While the intention is clear, the "Rescaling Prompt" shown in Figure 3 does not specify what the initial score is from which points should be deducted

- It is also not clear how information is measured, so a phrase like "half of the correct information" is left wide open to subjective interpretation

**Reasons To Accept:**

-

**Reasons To Reject:**

1. The proposed method is not motivated well, to the point that it is fundamentally unclear why and how it should address the research problem, for two reasons:

1.1 It is left unclear why a finer rating scale is better suited to capture the inherent subjectivity of the task at hand. The only motivation given is a reference to Kocmi & Federmann (2023), "which validated the effectiveness of a 0-100 scale compared to other options." What "effectiveness" means and if/how their results from machine translation apply to ratings of answers in QA tasks is unclear. If "subjectivity" in the task at hands means that different annotators have different views of what information is important in a document, then their ratings have no monotonic one-dimensional order and no amount of granularity can fix that.

1.2. According to the introduction, "subjectivity (and thus, inherent disagreement) in annotated data has surfaced as a key direction" and the authors make the observation that "intricacies in human judgments can be captured by natural language explanations provided during data annotation, which capture more nuanced subjectivity". So far so good, but now the authors claim that their "key idea is to make the ordinal label space (e.g., a coarse Likert scale) fine-grained, namely a 0-100 scale Kocmi & Federmann (2023), which then enables us to place the initial annotations in it by leveraging natural language explanations in a principled manner."
In what manner the proposed approach is "principled" is not clear to me. The authors are possibly referring to the scoring rubric that is included in the prompt, but just because some instruction is included in the prompt does not mean that the LM adheres to it, and it is even less clear if the LM faithfully can execute complex instructions that require it to repeatedly subtract a variable number of points, with the amount of points varying on the amount of information that is missing from the answer to be rated.

2. The benefit of the finer-grained ratings is not shown. The evaluation stops at showing moderate agreement with expert-produced fine-grained ratings and stating that the rescaling does not impact inter-annotator agreement. The introduction claimed that the proposed method can "capture more nuanced subjectivity", but this is never directly shown.

---

> ### Author Rebuttal · Authors · 2024-05-29
>
> Thank you for your review! Addressing your concerns/questions below:
>
> > In what manner the proposed approach is "principled" is not clear to me. [...] just because some instruction is included in the prompt does not mean that the LM adheres to it
>
> We agree! This was the intention behind our evaluation in Table 3, where we measure how the proposed rescaling compares to human rescaling (also referred to as reference scores in the paper). We measure the correlation and MAE between the human and the model rescaled scores to show that the model rescaling is faithful to the instruction.
>
> >  The benefit of the fine-grained ratings is not shown
>
> As we mentioned in response to reviewer kZfw as well, we don’t intend to claim that the full 100-point scale is needed. GPT-4 gives scores that are multiples of 5, so we aren’t using the full granularity of the scale. However, we *do* need more than a coarse 4-point Likert scale. We believe that our proposed method can work with roughly 10-20 score gradations. We will clarify this point in any future version.
>
> > "Rescaling Prompt" shown in Figure 3 does not specify what the initial score
>
> The rescaling prompt in Figure 3 mentions the scale on which the scoring needs to be done. In our analysis and looking through the MAEs, we see the LLM operating as expected by deducting points from 100,  which is why we didn’t need to explicitly state the initial score.
>
> > It is also not clear how information is measured..[..]..subjective interpretation
>
> The label description along with the specifics of missing information in the explanations enables the human annotator as well as GPT4 to measure how much information is marked as missing and apply the right deductions. This way of rescaling enables the scores to reflect the user’s measure of the information, so the same information can be weighed differently based on how the user expressed their judgment.

---

### Official Review · Reviewer_kZfw · 2024-05-12

**Rating:** 7
**Confidence:** 4
**Ethics Flag:** 1

**Summary:**

This paper proposes to use LMs to rescale human quality judgements by using explanations provided by the human annotators. The authors motivate the work by stressing the importance and value of human annotation in today's NLP landscape for model adaptation and evaluation, especially as models become more capable and automatic metrics are no longer reliable for increasingly subjective or complex tasks. The proposed automatic rescaling method can be used for a few purposes: to calibrate humans who have different interpretations of numeric rankings, to take a coarse-grained rating and make it more fine-grained (such as on a 100-point scale, as the authors propose to do), and/or to apply or specify an arbitrary rubric or criterion *after* the annotations are collected. The authors first collect a dataset of 12.6k high-quality human annotations of labels with explanations (5 per instance). They then test their rescaling method on a document-grounded non-factoid QA task where the goal is to rate "completeness" of a model-generated answer. They find that, despite the subjectivity of the task, their method can 1) rescale judgements while retaining the same level of human agreement as the original data annotations, and 2) closes some of the gap with expert rescaling judgements made using the new rubric, indicating that the method is at least partially successful.

**Questions To Authors:**

Questions/comments:
- Will you release your code & dataset upon publication?
- The motivation for using such a granular scale (1-100) could be made stronger. It appears from the given example and reported human MAE that a 1-10 or 1-20 scale would be sufficient and that the super granular scale may invite noisy disagreement (for both humans and machines). Could you explain this choice a bit more?
- Elaboration on "limited choice of models" mentioned above: Only one model (GPT4-0613) is studied for its ability to do the rescaling task, so its not clear how the correlation w/ human experts would change for other models and thus if this technique would be useful or valid for them, which is an important question for those doing non-API research. Similarly, only OpenAI API models (though 3 of them) are used for the initial question answering task.

Related work:
- there are some interesting papers in psychology literature on the Likert Scale and humans' varying interpretations or calibrations; may be good to cite.

Typos/misc:
- many of your citations are not in parentheses but should be (i.e., by using `\citep{}`).
- please make Figs 2,4 larger given extra space in a camera-ready version.
- abstract: what do you mean by "propose a method to rescale ordinal annotations *and explanations* using LLMs"? seems like it should say "using explanations"
- Fig 1: would be good to also show the 3 expert scores here to get a sense of variance and that there is no 100% agreed ground-truth.
- Seciton 6.2 last paragraph: here $M$ are the (unaveraged) individual reference scores, correct? It's a little confusing to double up on this $M$ notation.
- Important to mention that being able to use an arbitrary scoring metric for rescaling *after* annotations are collected relies on the explanations containing enough detail for that particular aspect to be able to do so.

**Reasons To Accept:**

- The paper is extremely clearly written and well-motivated. Related work coverage is comprehensive.
- The research question is quite timely. The proposed rescaling is an interesting use-case of LMs that could resolve a number of issues in human annotation for subjective tasks. I think the paper will have a good audience at this conference.
- The experiments are thorough and well-executed. Using a) LLMs to automate tasks and b) crowdsourcing can have a number of pitfalls if not assessed or evaluated rigorously, but the authors have done a good job in both of these areas, including elaborate quality control details in the Appendix.

**Reasons To Reject:**

- A few methodological details could be expounded on or are limited in scope. The main one is a limited choice of models investigated. I elaborated on these below in "questions/comments". Overall, I don't think that these issues affect the contribution and validity of the paper, just constrain its scope a bit.
- Ultimately, the proposed metric is not extremely strong. The main results (Table 3) support the inclusion of a rubric, but the avg. EBR baseline with a rubric is quite competitive, which may be support for the fact that the 1-100 Likert Scale is too fine-grained (as I've also touched on in "questions/comments" below). I'd appreciate if the authors could elaborate on their hypotheses for why this may be or their arguments against using the average approach.
- I assume the authors will publicly release their dataset with publication, but if not, that will severely limit the impact of the paper.

---

> ### Author Rebuttal · Authors · 2024-05-29
>
> Thanks for your detailed reviews and comments! Addressing some of your questions:
>
>
> > Will you release your code & dataset upon publication?
>
> Yes, we will be releasing the dataset and code!
>
> > The motivation for using such a granular scale (1-100) could be made stronger [...] Could you explain this choice a bit more?
>
> The 0-100 scale was a scale primarily chosen to be a natural prompting target for LLMs. We note that the models we used only produced scores that were multiples of 5, so in practice responses are mapped to 21 distinct values. We believe that a 1-10 or 1-20 scale could also work if our rubrics were defined in terms of these. The key element here is that the scale provides a bit more granularity than the original 4-point scale and enough granularity for the rubric to differentiate different responses.
>
> >  The main results (Table 3) support the inclusion of a rubric, but the avg. EBR baseline with a rubric is quite competitive
>
> While Avg EBR is comparable, it is still a four-point scale. It lacks the fine-grained aspect of evaluation which our approach enables us to achieve. EBR gives us the ability to tease out differences in human judgment and be more nuanced which otherwise is not possible with a four-point scale.
>
> We will make sure to update the paper with the points you mentioned in your review!

---

> ### Comment · Reviewer_kZfw · 2024-06-06
> **Response**
>
> Thanks for the response, and glad to hear you will be releasing the dataset!
>
> As for the motivation for the 0-100 scale: it is still not clear to me why such a large scale was selected, though I can see that this is not a crucial design choice. I'd encourage you to make this a little more clear in the paper (that a scale with so many intervals is not needed, and the same results could (likely?) be achieved with a scale with fewer intervals but no multiple-of-5 limitation).
>
> >While Avg EBR is comparable, it is still a four-point scale. It lacks the fine-grained aspect of evaluation which our approach enables us to achieve. EBR gives us the ability to tease out differences in human judgment and be more nuanced which otherwise is not possible with a four-point scale.
>
> I may be misunderstanding Table 3, but doesn't it follow from a comparable MAE between avg. EBR and EBR, meaning that each method's error in aligning with human judgement scores (0-100) is about the same, that either 1) a more fine-grained scale *does not* give the ability to tease out differences in human judgment, or 2) the human judgements do not have such nuances? I see nuanced human judgements in Figure 4a, so I assumed it was the former.
>
> I will keep my score.

---

### Decision · Program_Chairs · 2024-07-10

**Decision:**

Accept

**Comment:**

This paper uses language models (LMs) to rescale human quality judgments via annotator explanations. The method addresses the importance of human annotations for model adaptation and evaluation as models become more capable. It can calibrate differing human interpretations, refine coarse ratings, and apply rubrics post-annotation. Using a dataset of 12.6k annotations, the method was tested on a QA task, successfully rescaling judgments while maintaining agreement and aligning closer to expert evaluations.

Reviewers have mainly raised concerns about the validity of using a fine-grained 0-100 scale instead of a narrower range (e.g., 1-10).

In NLP, the 0-100 scale annotation has been primarily proposed and utilized in the machine translation community, supported by some advantages from a statistical point of view. It can capture subtle differences in the outputs, better aligning with human preference.
Relevant works worth mentioning in the paper include:
- Graham et al. (2013): "Continuous Measurement Scales in Human Evaluation of Machine Translation"
- Bojar et al. (2016): "Findings of the 2016 Conference on Machine Translation"
- Sakaguchi et al. (2018): "Efficient Online Scalar Annotation with Bounded Support"

We expect the authors to address other concerns and questions in the final draft.